# ROYAL SOCIETY
# OPEN SCIENCE

ecology/health and disease and epidemiology/theoretical biology

waterborne disease transmission, spatially explicit system, multi-layer network, reactivity, generalized stability theory

**Author for correspondence:**
Lorenzo Mari
e-mail: lorenzo.mari@polimi.it

# Conditions for transient epidemics of waterborne disease in spatially explicit systems

Lorenzo Mari[1], Renato Casagrandi[1], Enrico Bertuzzo[2], Andrea Rinaldo[3,4] and Marino Gatto[1]

[1]Dipartimento di Elettronica, Informazione e Bioingegneria, Politecnico di Milano, 20133 Milano, Italy
[2]Dipartimento di Scienze Ambientali, Informatica e Statistica, Università Ca' Foscari Venezia, 30170 Venezia Mestre, Italy
[3]Laboratory of Ecohydrology, Ecole Polytechnique Fédérale de Lausanne, 1015 Lausanne, Switzerland
[4]Dipartimento ICEA, Università di Padova, 35131 Padova, Italy

LM, 0000-0003-1326-9992; RC, 0000-0001-5177-803X;
EB, 0000-0001-5872-0666; AR, 0000-0002-2546-9548;
MG, 0000-0001-8063-9178

Waterborne diseases are a diverse family of infections transmitted through ingestion of—or contact with—water infested with pathogens. Outbreaks of waterborne infections often show well-defined spatial signatures that are typically linked to local eco-epidemiological conditions, water-mediated pathogen transport and human mobility. In this work, we apply a spatially explicit network model describing the transmission cycle of waterborne pathogens to determine invasion conditions in metacommunities endowed with a realistic spatial structure. Specifically, we aim to define conditions under which pathogens can temporarily colonize a set of human communities, thus triggering a transient epidemic outbreak. To that end, we apply generalized reactivity analysis, a recently developed methodological framework for the study of transient dynamics in ecological systems subject to external perturbations. The study of pathogen invasion is complemented by the detection of the spatial signatures associated with the perturbations to a disease-free system that are expected to be amplified the most over different time scales. Understanding the drivers of waterborne disease dynamics over time scales that are relevant to epidemic and/or endemic transmission is a crucial, cross-disciplinary challenge, as large portions of the developing world still struggle to cope with the burden of these infections.

# 1. Introduction

Waterborne diseases are infections caused by ingestion of (or, more generally, contact with) water contaminated by pathogenic organisms, ranging from micro- (protozoa, bacteria, viruses, algae) to macro-parasites (helminths like flatworms and roundworms). They still represent a major threat to human health, especially in low-income countries. Cholera and typhoid fever are among the best-known examples of potentially lethal waterborne diseases. Also, diarrhea, commonly associated with waterborne pathogens, is responsible for the deaths of about 1.5 million people every year, thus representing one of the leading causes of death, especially among infants and children in the developing world [1]. Unsafe water supply, lack of sanitation and poor hygienic conditions, which either directly or indirectly affect exposure and transmission rates, are crucial factors in determining the burden of waterborne infections [2].

Developing appropriate mechanistic tools is fundamental to understand, predict and control waterborne disease transmission. Typically, the development of such tools lies at the interface between mathematical biology, environmental science and epidemiology. A prototypical model for microparasitic waterborne infections was proposed by Codeço [3], who developed a system of three ordinary differential equations (ODEs) where, in addition to the compartments of susceptible ($S$) and infected ($I$) that characterize traditional models for microparasite transmission, one equation accounts for the population dynamics of pathogens ($B$) in the water reservoir used by the human community under study. Spatial dynamics were not accounted for in early studies on waterborne disease transmission. However, the spatial component of disease spread is at least as important as its temporal dynamics in determining local changes in the abundances of the aforementioned epidemiological compartments, especially in large-scale applications. The spatial spread of waterborne diseases is driven by both water-mediated transport [4] and human mobility [5]. The former typically occurs within a drainage basin, along natural or man-made network systems (i.e. rivers or canals, respectively), while the latter provides an efficient means of across-catchment (and possibly long-distance) pathogen dispersal. Therefore, spatially explicit modelling is crucial to understand how different spatial transmission mechanisms interact with each other and influence the spread of waterborne infections. It may also enable scientists and decision makers to reproduce with unprecedented accuracy real-world epidemiological dynamics (e.g. [6–10]), provide scenarios for future epidemic development over different time scales (e.g. [11–16]), and discuss the effectiveness of controls in real (e.g. [17–20]) and realistic (e.g. [21,22]) case studies. Spatial modelling also radically changed the traditional approach for the definition of conditions for pathogen establishment in complex applications, classically based on the evaluation of the basic reproduction number, $R_0$, defined as the average number of secondary infections caused by one infectious individual introduced in a completely susceptible community: in fact, it has been noted that values of $R_0$ locally greater than one are neither necessary nor sufficient for outbreaks to occur when spatial interactions are a factor, as processes like hydrologic transport and human mobility can radically alter local transmission dynamics [23–27].

While the study of the asymptotic properties of spatially explicit transmission models can help design effective control strategies intended to permanently reduce pathogen spread (or even break disease transmission), understanding transient phenomena of potential epidemiological interest can be important as well, namely to possibly prevent transitory epidemics triggered by external perturbations to a system in which endemic transmission is not possible—i.e. a system in which the so-called disease-free equilibrium (DFE) is asymptotically stable. Neubert & Caswell [28] proposed a simple measure of a system's short-term instability to small perturbations. Specifically, they defined *reactivity* as the maximum instantaneous rate at which perturbations to a stable steady state can be amplified. Although reactivity has been studied in several ecological contexts (see e.g. [29] and references therein), epidemiological applications are still relatively scarce: Hosack *et al.* [30] performed a complete reactivity analysis of Ross's [31] malaria model; Chitnis *et al.* [32] used reactivity analysis to derive epidemicity thresholds for simple models of Rift Valley fever transmission; Woodall *et al.* [33] applied reactivity analysis to a host–pathogen system with culling of the host population; Mari *et al.* [29] studied short-term instabilities connected to disease transmission in spatially implicit metapopulations and Mari *et al.* [34] studied the reactivity properties of simple (spatially implicit) models for waterborne and water-related diseases.

In this work, we use the recently developed method of generalized reactivity (hereafter, simply, g-reactivity), which is specifically tailored for ecological and epidemiological systems [29]. The definition

of g-reactivity is based on the analysis of the maximum amplification rate of external perturbations to a steady state as measured through a linear transformation of the state variables (the system output) that can be chosen so as to be epidemiologically relevant. This is achieved, in particular, by including all the infection-related components of the state space (such as the abundances of exposed or infected people, or the concentration of pathogens in an environmental reservoir) in the output, while at the same time excluding the others (such as the abundances of susceptible or recovered people; see [34]). Here, we extend the g-reactivity framework to encompass the analysis of waterborne disease dynamics in realistic, spatially explicit settings. Although reactivity has already been used in the analysis of spatially extended metapopulation models (e.g. for the study of Turing instabilities; see [35]), to the best of our knowledge the present paper represents one of the first attempts to study the reactivity properties of a realistic spatial system (see [36] for an application to marine protected areas), and the first to apply g-reactivity analysis in a spatially explicit context. Specifically, we aim to produce a complete characterization of the g-reactivity properties of the DFE of a model describing the transmission of waterborne microparasites in a metacommunity embedded in a realistic landscape. The analysis of transient dynamics is contrasted with the study of the asymptotic stability of the DFE (following [23,24]), to allow for a full comparison of the conditions leading to short- versus long-term waterborne pathogen invasion in spatially explicit systems.

The manuscript is organized as follows. In the next section, we outline a network model aimed to describe the transmission cycle of waterborne microparasites. Although general in its formulation, the processes described in the model are actually inspired by cholera transmission dynamics. In the following two sections, we derive spatially explicit conditions under which the DFE of the network system is g-reactive (so that suitable perturbations to the DFE can be temporarily amplified in an epidemiologically relevant system output, before eventually fading out) or unstable (so that pathogens can invade the community and establish permanently therein). Next, we study the geographical signatures of the largest-growing perturbations to the DFE, as well as the geography of epidemic outbreak. Outbreak and establishment conditions are then numerically analysed in spatially explicit systems, each consisting of a set of human settlements distributed along a realistic river network. Finally, a short discussion of the main results of the work, with a focus on eco-epidemiological implications, closes the paper.

## 2. The model

The study of waterborne pathogen transmission is here tackled by means of a network model [10] that has already been used as starting point for both theoretical studies on waterborne disease dynamics [5,24,26] and the analysis of real-world cholera epidemics [11–16,19,23]. The model describes local epidemiological, demographic and ecological processes, pathogen transport along water systems and the effects of short-term human mobility on disease propagation. Network nodes represent $n$ human communities of assigned population size, arranged in a given spatial setting, and connected by hydrologic pathways and human mobility.

Let $S_i(t)$ and $I_i(t)$ be the local abundances of susceptible and infected individuals in each node $i$ of the network at time $t$, and let $B_i(t)$ be the concentration of pathogens in the local water reservoirs which human communities have access to. Epidemiological dynamics and pathogen transport over the hydrologic and human mobility networks can be described by the following set of $3n$ ODEs:

$$\frac{dS_i}{dt} = \mu(N_i - S_i) - \left[ (1 - m_i^S)\beta_i \frac{B_i}{K + B_i} + m_i^S \sum_{j=1}^{n} Q_{ij}\beta_j \frac{B_j}{K + B_j} \right] S_i,$$

$$\frac{dI_i}{dt} = \left[ (1 - m_i^S)\beta_i \frac{B_i}{K + B_i} + m_i^S \sum_{j=1}^{n} Q_{ij}\beta_j \frac{B_j}{K + B_j} \right] S_i - (\mu + \delta + \gamma)I_i$$

and

$$\frac{dB_i}{dt} = -(\nu_i + l_i)B_i + \frac{1}{W_i} \sum_{j=1}^{n} l_j P_{ji} W_j B_j + \frac{p_i}{W_i} \left[ (1 - m_i^I)I_i + \sum_{j=1}^{n} m_j^I Q_{ji} I_j \right]. \tag{2.1}$$

As for the human host population, the dynamics of the susceptible compartment in each community (first equation of model (2.1)) is described as a balance between population demography and infections due to exposure to the pathogen. The host population, if uninfected, is assumed to be at demographic equilibrium $N_i$, with $\mu$ being the human mortality rate. The parameter $\beta_i$ represents the site-specific

rate of exposure to contaminated water, and $B_i/(K + B_i)$ is the dose–response function describing the probability of becoming infected due to the exposure to a concentration $B_i$ of pathogens (with $K$ being the half-saturation constant [3]). Exposure to contaminated water for susceptible people of community $i$ can occur either in their home community (with probability $1 - m_i^S$, with $m_i^S$ being the overall probability of exposure outside the home site $i$, as determined by the mobility of susceptible individuals) or elsewhere (with probability $m_i^S Q_{ij}$, with $Q_{ij}$ representing the probability that water contacts taking place outside the home site $i$ occur in site $j \neq i$, $Q_{ii} = 0$). Other routes of infections, such as fast human-to-human transmission, which have been proposed for waterborne diseases like cholera, are here neglected for simplicity, but could be dealt with within the same modelling framework (e.g. [27]).

The evolution of the infected compartment (second equation of model (2.1)) is a balance between newly infected individuals and losses due to recovery or natural/pathogen-induced mortality, with $\delta$ and $\gamma$ being the rates of disease-induced mortality and recovery from infection, respectively. Note that the recovered compartment is not modelled explicitly, so that individuals who recover from the acute phase of disease are simply removed from the population, as they were conferred life-long immunity to reinfection. For cholera, as an example, recent estimates place the duration of immunity in the range 2.3–3.0 years [13]. As such, loss of acquired immunity is unlikely to influence transient, short-term epidemic dynamics, which are the main focus of the present work. Although simplistic, the choice of neglecting the dynamics of recovered individuals is thus deemed reasonable for the problem at hand.

As for the pathogen population, the dynamics of the local concentrations of pathogens in the aquatic environment (third equation of model (2.1)) is given by a balance between water contamination, pathogen mortality and hydrologic transport. Pathogens are released in water (e.g. excreted) by infected individuals (from either the local community, with probability $1 - m_i^I$, with $m_i^I$ being the overall probability of contamination outside the home site $i$, as determined by the mobility of infected individuals; or elsewhere, with probability $m_j^I Q_{ji}$ at a site-specific rate $p_i$ and immediately diluted in a well-mixed local water reservoir of size $W_i$. Free-living pathogens are assumed to die at rate $\nu_i$. They can also move between any two neighbouring nodes of the hydrologic network (say from $i$ to $j$) at rate $l_i$ and with probability $P_{ij}$.

Some of the parameters of model (2.1)—namely those related to human demography ($\mu$) and the physiological response to the disease ($\delta$, $\gamma$, $K$)—are assumed to be constant over the spatial scales considered in this study. All the other parameters are allowed to be possibly site-dependent. A summary of the state variables and parameters of model (2.1) is given in table 1. We finally note that setting $m_i^S = m_i^I = 0$ and $l_i = 0$ for all $i$'s in model (2.1) produces a set of $n$ disconnected models accounting only for local disease transmission processes. Unless stated otherwise, all the analyses and results presented in the next sections refer to the full model accounting also for spatial coupling mechanisms.

# 3. Conditions for short-term pathogen outbreak

To study short-term pathogen outbreaks, we seek conditions under which small perturbations to the DFE, the state of model (2.1) in which $S_i = N_i$, $I_i = 0$ and $B_i = 0$ for all $i$'s, can initially grow. If the DFE is unstable, that is if the generalized reproduction number $\mathcal{R}_0$ is larger than one (note that $\mathcal{R}_0$ can be worked out from the analysis of the Jacobian $\mathbf{J_0}$, i.e. the matrix that describes the dynamics of the system linearized in a neighbourhood of the DFE; for details, see appendix A in the electronic supplementary material), virtually any perturbation can grow and generate an epidemic, which is eventually followed by the establishment of endemic pathogen transmission. More subtle is the case of a stable DFE ($\mathcal{R}_0 < 1$), in which perturbations can either decay monotonically or undergo transient growth (thus possibly generating an epidemic wave) before eventually fading out. Neubert & Caswell [28] proposed a simple, yet extremely effective method to characterize the transient dynamics associated with a stable equilibrium of a linear (or linearized) system of ODEs after a pulse perturbation. Specifically, they defined as *reactive* those stable steady states for which there exist small perturbations that can yield a transient amplification of the Euclidean norm of the state vector. This definition has recently been extended to a generalized, fully anisotropic reactivity framework (g-reactivity, [29]).

Following Mari *et al.* [29], the DFE of model (2.1) is g-reactive if there exist small perturbations that can lead to a transient growth of the Euclidean norm of a suitable system output (**y**) that is linked to the

**Table 1.** State variables and parameters of model (2.1), and parameters of the output transformation (3.2).

| symbol | definition |
| --- | --- |
| | *state variables* |
| $S_i$ | abundance of susceptible human hosts at site $i$ ($i = 1 \ldots n$) |
| $I_i$ | abundance of infected human hosts at site $i$ |
| $B_i$ | concentration of pathogens in the water reservoirs of site $i$ |
| | *parameters: local processes* |
| $N_i$ | human population size in absence of disease at site $i$ |
| $\mu$ | baseline human mortality rate (site-independent) |
| $\beta_i$ | rate of exposure to contaminated water at site $i$ |
| $K$ | half-saturation constant of dose-response function (site-independent) |
| $\delta$ | disease-induced mortality rate (site-independent) |
| $\gamma$ | recovery rate (site-independent) |
| $p_i$ [$\theta_i$] | rate of water contamination at site $i$ [$\theta_i = p_i/K$, rescaled contamination rate] |
| $W_i$ | size of the water reservoir of site $i$ |
| $\nu_i$ | pathogen mortality rate at site $i$ |
| | *parameters: spatial coupling mechanisms* |
| $m_i^S$ | overall probability of exposure outside home site $i$ |
| $m_i^I$ | overall probability of contamination outside home site $i$ |
| $Q_{ij}$ | probability of water contact at $j$ conditional to occurring outside home site $i$ |
| $l_i$ | hydrologic transport rate of pathogens at site $i$ |
| $P_{ij}$ | probability of hydrologic pathogen transport between sites $i$ and $j$ |
| | *parameters: output transformation* |
| $c_{Ii}$ | weight assigned to infected hosts at site $i$ |
| $c_{Bi}$ | weight assigned to bacterial concentration at site $i$ |

full state of the system ($\mathbf{x} = [\mathbf{s}^{\mathrm{T}}, \mathbf{i}^{\mathrm{T}}, \mathbf{b}^{\mathrm{T}}]^{\mathrm{T}}$, a $3n$-dimensional vector whose components $\mathbf{s} = [S_1, \ldots, S_n]^{\mathrm{T}}$, $\mathbf{i} = [I_1, \ldots, I_n]^{\mathrm{T}}$ and $\mathbf{b} = [B_1/K, \ldots, B_n/K]^{\mathrm{T}}$ correspond to susceptible humans, infected humans and bacterial concentrations (properly rescaled); the superscript T indicates matrix transposition) through a linear transformation ($\mathbf{y} = \mathbf{Cx}$, where $\mathbf{C}$ is a full-rank $q \times 3n$ real matrix, $q \leq 3n$; hence $\mathbf{y}$ is a $q$-dimensional vector). In other words, the DFE is g-reactive if

$$\left. \frac{\mathrm{d}\|\mathbf{y}\|}{\mathrm{d}t} \right|_{t=0} > 0, \tag{3.1}$$

at least for some perturbations $\mathbf{x_0} = \mathbf{x}(0)$, which identify the so-called g-reactivity basin of the equilibrium. In epidemiological applications, inequality (3.1) also corresponds to the condition for the possible occurrence of transient epidemic waves, contingent upon a suitable choice for matrix $\mathbf{C}$: specifically, the output matrix should include all and only the infection-related states of the system [34]. For the sake of generality, here we define matrix $\mathbf{C}$ as

$$\mathbf{C} = \begin{bmatrix} \mathbf{0_n} & \mathbf{c_I} & \mathbf{0_n} \\ \mathbf{0_n} & \mathbf{0_n} & \mathbf{c_B} \end{bmatrix}, \tag{3.2}$$

where $\mathbf{0_n}$ is the null matrix of dimension $n$; and $\mathbf{c_I}$ and $\mathbf{c_B}$ are diagonal matrices with positive elements representing the weights given to the infected and bacterial components of the state space in the output transformation, respectively. We thus have $\mathbf{y} = [c_{I1} I_1, \ldots, c_{In} I_n, c_{B1} B_1/K, \ldots, c_{Bn} B_n/K]^{\mathrm{T}}$. Note that temporary fluctuations of the susceptible compartment cannot directly influence the g-reactivity properties of the DFE, because susceptible human hosts do not contribute to the system output.

Using definition (3.1) and the output transformation matrix $\mathbf{C}$ from (3.2), it is possible to show (appendix B, electronic supplementary material) that the DFE of model (2.1) is g-reactive if

$$\lambda_{\max}(\mathbf{H_0}) > 0, \tag{3.3}$$

where $\lambda_{\max}$ $(\mathbf{H_0})$ indicates the dominant eigenvalues of matrix $\mathbf{H_0} = H(\mathbf{CJ_0C^+}) = (1/2)(\mathbf{CJ_0C^+} + (\mathbf{C^+})^{\mathrm{T}}\mathbf{J_0^T}\mathbf{C^T})$, that is the Hermitian part of matrix $\mathbf{J_0}\mathbf{C^+}$, with $\mathbf{C^+} = \mathbf{C^T}(\mathbf{CC^T})^{-1}$ being the right pseudo-inverse of matrix $\mathbf{C}$. Inequality (3.3) represents the necessary—yet not sufficient—condition for the occurrence of a (short-term) epidemic outbreak. In fact, transient waves of infections may actually occur only for perturbations lying within the g-reactivity basin of the DFE. A MATLAB$^{\mathrm{TM}}$ implementation of the instructions required to evaluate the stability and g-reactivity properties of the DFE of model (2.1) is provided in appendix C (electronic supplementary material).

It is also worth noticing that the g-reactivity properties of the DFE can actually be evaluated based on a matrix of reduced order $n$, namely $\mathbf{F_0} = \mathbf{T_0} + \mathbf{E_0} + \mathbf{E_0^S} + \mathbf{E_0^I} + \mathbf{E_0^{SI}}$ (details in appendix B), where

$$\mathbf{T_0} = \boldsymbol{\nu}^{-1}(\mathbf{c_B}\mathbf{W}^{-1}\mathbf{P^T}\mathbf{W}\mathbf{c_B}^{-1} - \mathbf{U_n})\mathbf{l},$$

$$\mathbf{E_0} = \frac{\boldsymbol{\nu}^{-1}}{4\phi}[\mathbf{c_I}(\mathbf{U_n} - \mathbf{m^S})\mathbf{N}\boldsymbol{\beta}\mathbf{c_B}^{-1} + \mathbf{c_B}\boldsymbol{\theta}\mathbf{W}^{-1}(\mathbf{U_n} - \mathbf{m^I})\mathbf{c_I}^{-1}]^2,$$

$$\mathbf{E_0^{SI}} = \frac{\boldsymbol{\nu}^{-1}}{4\phi}[\mathbf{c_I}\mathbf{m^S}\mathbf{N}\mathbf{Q}\boldsymbol{\beta}\mathbf{c_B}^{-1} + \mathbf{c_B}\boldsymbol{\theta}\mathbf{W}^{-1}\mathbf{Q^T}\mathbf{m^I}\mathbf{c_I}^{-1}]^2,$$

$$\mathbf{E_0^S} = \frac{\boldsymbol{\nu}^{-1}}{2\phi}[\mathbf{c_I}(\mathbf{U_n} - \mathbf{m^S})\mathbf{N}\boldsymbol{\beta}\mathbf{c_B}^{-1} + \mathbf{c_B}\boldsymbol{\theta}\mathbf{W}^{-1}(\mathbf{U_n} - \mathbf{m^I})\mathbf{c_I}^{-1}]\mathbf{c_I}\mathbf{m^S}\mathbf{N}\mathbf{Q}\boldsymbol{\beta}\mathbf{c_B}^{-1}$$

and

$$\mathbf{E_0^I} = \frac{\boldsymbol{\nu}^{-1}}{2\phi}[\mathbf{c_I}(\mathbf{U_n} - \mathbf{m^S})\mathbf{N}\boldsymbol{\beta}\mathbf{c_B}^{-1} + \mathbf{c_B}\boldsymbol{\theta}\mathbf{W}^{-1}(\mathbf{U_n} - \mathbf{m^I})\mathbf{c_I}^{-1}]\mathbf{c_B}\boldsymbol{\theta}\mathbf{W}^{-1}\mathbf{Q^T}\mathbf{m^I}\mathbf{c_I}^{-1},$$

correspond to the contribution of hydrologic pathogen transport ($\mathbf{T_0}$), locally occurring exposure and contamination ($\mathbf{E_0}$), mobility-driven exposure and contamination ($\mathbf{E_0^{SI}}$), and a mixture of local and mobility-driven processes ($\mathbf{E_0^S}$ and $\mathbf{E_0^I}$). In the above matrix definitions, $\mathbf{U_n}$ is the identity matrix of dimension $n$; $\phi = \mu + \delta + \gamma$; and $\mathbf{N}$, $\mathbf{W}$, $\boldsymbol{\beta}$, $\boldsymbol{\theta}$, $\boldsymbol{\nu}$, $\mathbf{m^S}$, $\mathbf{m^I}$ and $\mathbf{l}$ are diagonal matrices with positive entries corresponding to the parameters $N_i$, $W_i$, $\beta_i$, $\theta_i = p_i/K$, $\nu_i$, $m_i^S$, $m_i^I$ and $l_i$, with $i = 1, \ldots, n$. The transient epidemicity condition can thus equivalently be stated as

$$\mathcal{E}_0 = \lambda_{\max}(\mathbf{F_0}) > 1. \tag{3.4}$$

Condition (3.4) parallels the asymptotic stability criterion based on the generalized reproduction number $\mathcal{R}_0$ (appendix A). Together, they generalize g-reactivity and stability analysis results found in spatially implicit applications [34] to spatial settings of any complexity.

# 4. Identification of critical perturbations over different time scales

The analysis of the dominant eigenvalue of matrix $\mathbf{H_0}$ (or $\mathbf{F_0}$) enables us to determine whether small perturbations to a stable DFE can be temporarily amplified in a suitable (i.e. epidemiologically relevant) system output. By the Rayleigh principle [37], the eigenvector associated with the dominant eigenvalue of $\mathbf{H_0}$ corresponds to the structure of the single perturbation (lying in the row space of $\mathbf{C}$) characterized by the largest rate of amplification in the limit $t \to 0$, i.e. the optimal perturbation at time 0 [28,38]. According to the Perron–Frobenius theorem for non-negative matrices [37], the dominant eigenvector of $\mathbf{H_0}$ is characterized by strictly positive components. Because model (2.1) is endowed with a well-defined spatial structure, the dominant eigenvector of matrix $\mathbf{H_0}$ can be interpreted as the geographic signature of the perturbation leading to the fastest outbreak in the short term. On the other hand, it is possible to show (see again appendix B) that the dominant eigenvector of $\mathbf{F_0}$ refers to the bacterial components of the state space, although only close to the transient epidemicity boundary $\mathcal{E}_0 = 1$.

One could also wonder whether there exist ways to determine the geographic signature of the perturbations leading to the largest epidemic amplification (measured as $\alpha_y(t) = ||\mathbf{y}(t)||/||\mathbf{y}(0)||$) over longer time scales. In this respect, generalized stability theory (see again [38]) predicts that, in a linear (or linearized) system, the perturbation leading to the largest growth of the Euclidean norm of the system state (say, $\alpha_x(t) = ||\mathbf{x}(t)||/||\mathbf{x}(0)||$) for $t \to \infty$ can be obtained from the eigenvectors of the Jacobian matrix of the system, specifically as the conjugate of the biorthogonal

of the leading eigenvector of the Jacobian. Going back to our problem, we know (appendix A) that $J_0$ is block-triangular, with the infection-related components of model (2.1), namely infected people and free-living pathogens, basically constituting an isolated sub-system that is not influenced by the susceptible components of the model. The dynamics of this subsystem close to the DFE are thus completely determined by $J'_0$, that is the submatrix of the Jacobian $J_0$ of the full system restricted to the infection-related state variables. Recalling that, with the definition of matrix $C$ given in (3.2), susceptibles are assumed not to influence the system output, it can immediately be verified that $H_0 = H(J'_0) = (J'_0 + J'^T_0)/2$ if $c_I$ and $c_B$ are both equal to the identity matrix of size $n$, $U_n$. In this case, in fact, $\alpha_y(t)$ corresponds to $\alpha_x(t)$ evaluated for the infection-related subsystem (in which $x = [i^T, b^T]^T$). Therefore, we can conclude that the largest-growing perturbation in the system output for $t \to \infty$ is given by the conjugate of the biorthogonal of the leading eigenvector of $J'_0$, albeit only if $c_I = c_B = U_n$.

We finally remark that the asymptotic results drawn from generalized stability theory can provide meaningful indications about the behaviour of a nonlinear system around an equilibrium point only if the associated Jacobian can describe the dynamics of the system over relatively long time scales. In the problem at hand, this may prove true for a stable DFE, in particular for parameter combinations for which the initial depletion of the susceptible compartment is not too fast. If this is the case, then the asymptotic optimal perturbation based on the eigenvectors of matrix $J'_0$ may represent a heuristic upper bound to the long-term amplification of generic small perturbations to a stable DFE.

# 5. Outbreak and establishment conditions in a river network

## 5.1. Application to realistic settings

To analyse outbreak/establishment conditions in a realistic setting, we assume that human communities constitute the nodes of a so-called optimal channel network (OCN), i.e. a mathematical structure characterized by scaling forms that closely conform to the observed geomorphological features of real river networks [39–42]. Because of this feature, OCNs have often been used as a template for the structure of the landscape in epidemiological applications (e.g. [4,5,7,24,26,43]). We further assume, without loss of generality, that the OCN landscape is embedded in a square of unitary side (this condition can be relaxed by imposing periodic boundary conditions). We consider OCNs endowed with three different spatial configurations, characterized by the position of the outlet (figure 1). OCNs have been generated following the algorithm described in Bertuzzo *et al.* [44,45], based on the procedure proposed by Rigon *et al.* [46]. Since the generation of OCNs is an intrinsically stochastic process, we consider several (16) replicas for each network geometry. The total number of network nodes ($n \approx 500$) is preserved in each geometry and replica.

As for water-mediated pathogen movement, we apply conservative transport everywhere except for the outlet, from which pathogens are removed from the network. The specification of matrix $P = [P_{ij}]$ thus goes as follows. Let $p^d$ be the probability of downstream transport (and $1 - p^d$ the probability of upstream transport). If $h$ is a headwater node and $j$ its downstream neighbour, $P_{hj} = 1$ (reflecting boundary); for the inner nodes $i$ of the network, $P_{ij} = p^d$ if $j$ is the downstream neighbour, or $P_{ij} = (1 - p^d)/n^u_i$ if $j$ is one of the $n^u_i$ upstream neighbours of node $i$, with $n^u_i = 2$ for most $i$'s; at the outlet, $o$, pathogens are discharged from the network with probability $p^d$ (absorbing boundary), while $P_{oj} = (1 - p^d)/n^u_o$ if $j$ is one of the $n^u_o$ upstream neighbours of node $o$. Note that absorbing conditions prevent pathogens from accumulating at the network outlet.

On top of hydrologic connectivity, a second mobility layer accounting for human movement has also to be specified in model (2.1). To that end, pairwise movement probabilities $Q_{ij}$ are described through a gravity model [47,48] in which the attractiveness of node $j$ for node $i$ is assumed to be directly proportional to the population size of $j$ and inversely proportional to the distance $d_{ij}$ between the two nodes (through an exponential kernel with scale factor $D$), i.e. $Q_{ij} \propto N_j \exp(-d_{ij}/D)$ (if $i \neq j$, $Q_{ii} = 0$). Movement probabilities $Q_{ij}$ constitute the entries of the human mobility matrix ($Q = [Q_{ij}]$). To stipulate that $Q$ is row-stochastic (i.e. a matrix in which rows sum up to one), outgoing mobility fluxes are normalized by $\sum_{k \neq i}^n N_k \exp(-d_{ik}/D)$. Different mobility models can be easily accommodated in the formalism of system (2.1), provided that human mobility may be expressed in terms of movement probability (as quantified by $m^S_i$ and $m^I_i$) and trip distribution (e.g. in the form of an origin-destination matrix, as quantified by $Q$).

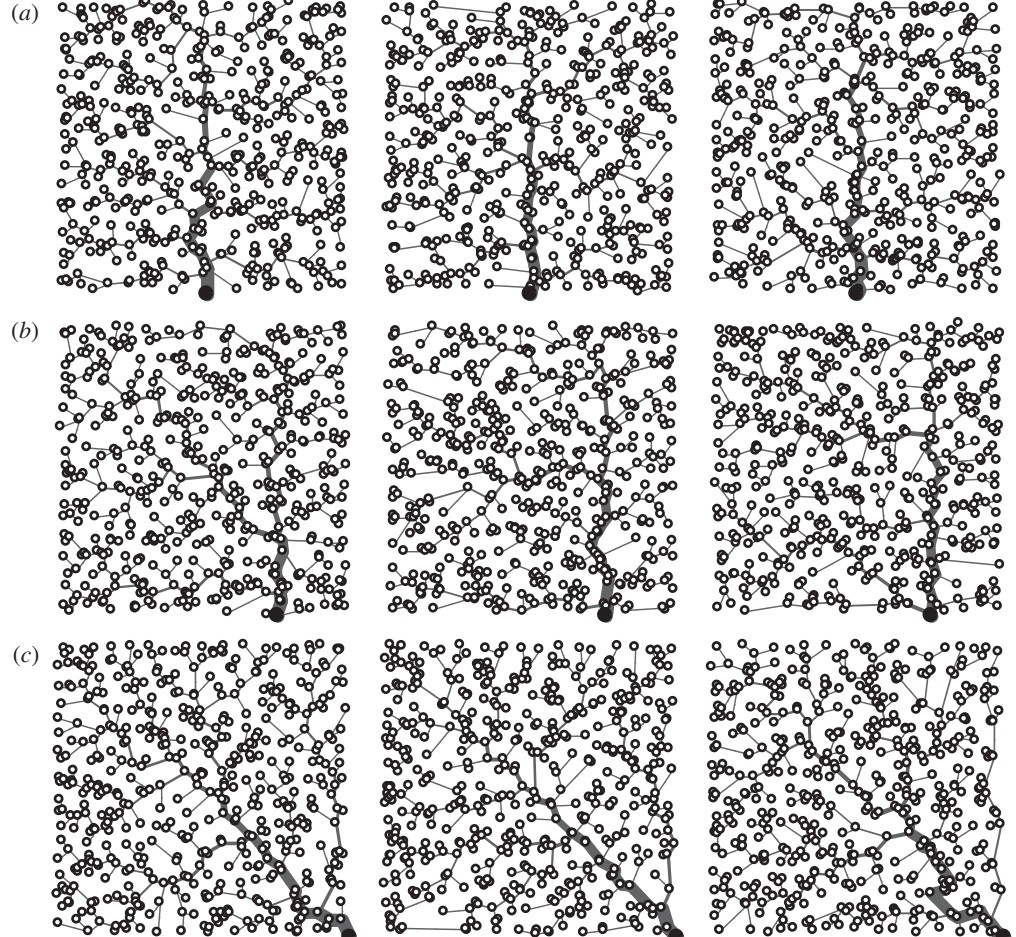

**Figure 1.** Examples of OCN topologies characterized by different positions of the outlet. (*a*) The network outlet (larger black node) is located at the mid-point of the bottom side of the domain. (*b*) The network outlet is located half-way between the mid-point and the right end-point of the bottom side of the domain. (*c*) The network outlet is located at the right end-point of the bottom side of the domain. Three different replicas (out of 16 considered overall, see text) are representatively shown for each of the three topologies.

## 5.2. Numerical analysis of threshold conditions

Figure 2*a* shows the g-reactivity and stability properties of the DFE of model (2.1) as a function of the human exposure and contamination rates ($\beta$ and $\theta = p/K$, respectively) when all model parameters, including the weights given to infected prevalence and bacterial abundance in the different communities, are assumed to be homogeneously distributed in space (therefore, $\mathbf{N} = N\mathbf{U_n}$, $\mathbf{W} = W\mathbf{U_n}$, $\boldsymbol{\beta} = \beta\mathbf{U_n}$, $\boldsymbol{\theta} = \theta\mathbf{U_n}$, $\boldsymbol{\nu} = \nu\mathbf{U_n}$, $\mathbf{m^S} = m^S\mathbf{U_n}$, $\mathbf{m^I} = m^I\mathbf{U_n}$, $\mathbf{l} = l\mathbf{U_n}$, $\mathbf{c_I} = c_I\mathbf{U_n}$ and $\mathbf{c_B} = c_B\mathbf{U_n}$ are all scalar matrices). Results refer to the leftmost OCN replica shown in figure 1*a*, but the g-reactivity and stability thresholds evaluated with all the other OCN structures are virtually indistinguishable from those of figure 2*a*. This shows that, for an assigned specification of hydrologic transport and human mobility, the g-reactivity and stability properties of model (2.1) may be quite robust to variations of the underlying spatial domain, concerning e.g. the fine-scale structure of the OCN landscape or the position of the outlet node.

If both the exposure rate $\beta$ and the contamination rate $\theta$ are small in magnitude, the DFE is stable ($\mathcal{R}_0 < 1$)—which prevents long-term pathogen establishment—and non-g-reactive ($\mathcal{E}_0 < 1$, evaluated for $c_I = 1$ and $c_B = 1$)—which rules out even transient epidemic outbreaks in the community. The parameter regions characterized by either relatively large values of $\beta$ and small values of $\theta$, or relatively large values of $\theta$ and small values of $\beta$ correspond instead to a stable, g-reactive DFE ($\mathcal{R}_0 < 1$, $\mathcal{E}_0 > 1$). In this case, transient epidemic waves can be triggered by suitable perturbations to the DFE, provided that either exposure or contamination is sufficiently large, but long-term pathogen establishment and endemic transmission are not possible. For these to happen, both $\beta$ and $\theta$ need to be sufficiently large.

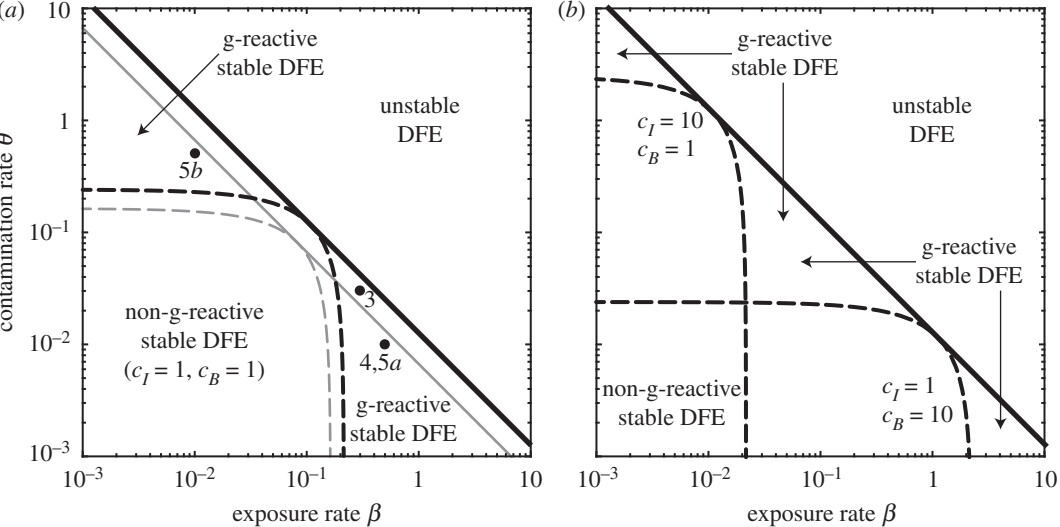

**Figure 2.** Stability and g-reactivity analysis for the DFE of model (2.1). (*a*) The solid black line ($\mathcal{R}_0 = 1$) separates the parameter combinations for which the DFE is stable ($\mathcal{R}_0 < 1$) from those for which the DFE is unstable ($\mathcal{R}_0 > 1$), while the dashed black line ($\mathcal{E}_0 = 1$) marks the boundary between the parameter sets for which the DFE is g-reactive ($\mathcal{E}_0 > 1$) and those for which the DFE is non-g-reactive ($\mathcal{E}_0 < 1$). These stability and g-reactivity thresholds refer to the leftmost OCN shown in figure 1*b*, but similar results were obtained with all of the other OCNs tested. The stability/g-reactivity boundaries for a local model (i.e. a model describing a set of disconnected local communities obtained from (2.1) by setting $l = 0$ and $m^S = m^I = 0$) are shown as thin grey lines: specifically, the local DFE is unstable above the solid line and non-g-reactive below the dashed line. The dots indicate some of the parameter combinations explored in figures 3–5. (*b*) G-reactivity analysis for different choices of the entries of matrix **C** (equation (3.2)). Parameter values [11]: $\mu = 4.2 \times 10^{-5}$, $\delta = 4.0 \times 10^{-4}$, $\gamma = 1/5$, $\nu = 1/30$, $l = 1/3$ (all rates in [day$^{-1}$]), $p^d = 0.8$, $m^S = 0.2$, $m^I = 0.05$, $D = 0.05$ [–]. All parameters are assumed to be spatially homogeneous, including the size of local communities ($N_i = N = 1$), the distribution of water resources ($W_i = W = 1$), and the weights of infected humans and bacterial concentrations in the output transformation ($c_{Ii} = c_I$ and $c_{Bi} = c_B$, respectively). See table 1 for a summary of model parameters.

We also note from figure 2*a* that higher values of $\beta$ and $\theta$ are needed in the network model (which includes pathogen transport and human mobility) than in a spatially implicit setting (i.e. in a model with $l_i = 0$ and $m_i^S = m_i^I = 0$ for all $i$'s) for the DFE to be g-reactive/unstable. This is specifically due to the presence of an absorbing boundary for hydrologic pathogen transport, which makes this spatial coupling mechanism non-conservative. As a matter of fact, if hydrologic transport of pathogens is negligible—or otherwise dominated by human mobility—a different result can be found, namely that epidemic/endemic transmission requires lower values of $\beta$ and $\theta$ in a spatially explicit (rather than in a spatially implicit) setting (e.g. [24]).

The definition of matrix **C** given in (3.2) clearly influences the g-reactivity properties of the DFE. In fact, choosing different values for $c_I$ or $c_B$ can imply a change in the classification of the DFE from g-reactive to non-g-reactive, or vice versa (figure 2*b*). Therefore, g-reactivity classification is not absolute: in order to be meaningful, it requires a suitable (i.e. epidemiologically motivated) design of the output transformation.

The role played by the transport/mobility parameters in triggering disease epidemicity or endemicity is shown in figure 3. High values of the hydrologic transport rate $l$ and the downstream transport probability $p^d$ are associated with a stable, non-g-reactive DFE, while transient epidemics and endemic transmission can be found for lower values of $l$ and/or $p^d$ (panel *a*). On the other hand, human mobility promotes both short-term outbreaks and long-term pathogen establishment (panel *b*). In fact, for low levels of human mobility (small $m^S$ and $m^I$) the pathogen cannot invade the system and the DFE is non-g-reactive. Finally, it is possible to study the interplay between water-mediated pathogen transport and human mobility (panel *c*). Again, high values of the hydrologic transport rate are associated with a stable, possibly non-g-reactive DFE. Conversely, high human mobility can lead to a g-reactive or unstable DFE. This general picture remains qualitatively unchanged for different values of the baseline exposure and contamination rates, although quantitative details can obviously vary. Interestingly, when looking at the parameters concerning spatial coupling mechanisms, we find that network topology may indeed influence the g-reactivity and stability properties of the DFE of model

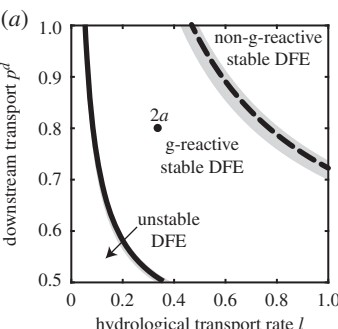
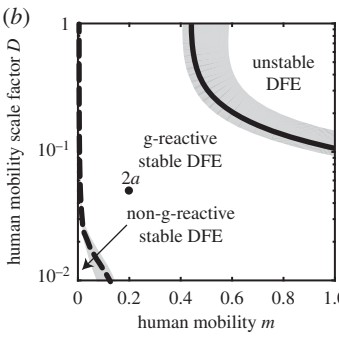
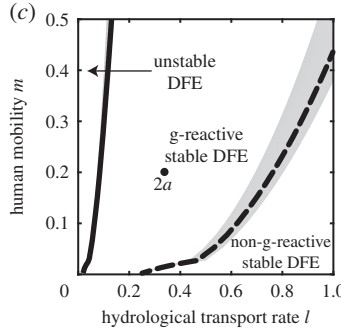

**Figure 3.** The role of spatial coupling mechanisms in disease epidemicity and endemicity. (*a*) Hydrologic transport of pathogens. (*b*) Human mobility. (*c*) Coupled effect of hydrologic pathogen transport and human mobility. Results are reported for 16 OCN replicas like those shown in figure 1*b*. Specifically, in each panel, the solid (dashed) black line represents the median location of the replica-specific stability ($\mathcal{R}_0 = 1$) [g-reactivity ($\mathcal{E}_0 = 1$)] boundary, while the grey-shaded areas are the envelopes of the boundaries obtained from each OCN replica. Dots indicate the combination of parameters related to hydrologic transport and/or human mobility used in figure 2*a*. Parameter values: $\beta = 0.3$, $\theta = 0.03$, $m^S = m$, $m^I = m/4$. Other parameters as in figure 2*a*. Results obtained for different OCN configurations like those shown in panels *a* and *c* of figure 1 are reported in figure S1, available as electronic supplementary material.

(2.1), especially in parameter regions characterized by relatively high hydrologic transport and/or human mobility (electronic supplementary material, figure S1).

## 5.3. Amplification of small perturbations to the DFE over different timescales

To verify that the analysis of the dominant eigenvector of matrix $\mathbf{H_0}$ does indeed make it possible to identify perturbations that are critical in terms of their initial rate of amplification, we solve numerically system (2.1) with different initial conditions and different OCN configurations. Specifically, we consider small perturbations in the form

$$\mathbf{w_0} = (1 - \sigma)\mathbf{w_{opt}} + \sigma\mathbf{w_{rand}} = (1 - \sigma)\begin{bmatrix} \mathbf{c_I^{-1}i_{opt}} \\ \mathbf{c_B^{-1}b_{opt}} \end{bmatrix} + \sigma\begin{bmatrix} \mathbf{i_{rand}} \\ \mathbf{b_{rand}} \end{bmatrix},$$

in which $\mathbf{w_{opt}}$ is proportional to the optimal perturbation at time 0 (i.e. the dominant eigenvector of $\mathbf{H_0}$; equivalently, $\mathbf{i_{opt}}$ and $\mathbf{b_{opt}}$ can be worked out from the dominant eigenvector of matrix $\mathbf{F_0}$ if $\mathcal{E}_0 \approx 1$) and $\mathbf{w_{rand}}$ is a spatially distributed random perturbation (with $\mathbf{i_{rand}}$ and $\mathbf{b_{rand}}$ being two independent random vectors of length $n$ drawn from a uniform distribution with non-negative support). The vectors $\mathbf{w_{opt}}$ and $\mathbf{w_{rand}}$ are suitably rescaled so that they have the same norm. Coherently with the definition of matrix $\mathbf{C}$ given in (3.2), the susceptible components of the state space are not perturbed ($S_i(0) = N_i$ for all $i$'s). The initial growth rate $\rho_0$ of a given perturbation is then evaluated numerically as

$$\rho_0 = \left. \left( \frac{1}{||\mathbf{y}||} \frac{d||\mathbf{y}||}{dt} \right) \right|_{t=0}.$$

Note that, by definition, the initial growth rate of the optimal perturbation at time 0 corresponds to the dominant eigenvalue of matrix $\mathbf{H_0}$ [28,29]. Figure 4*a* shows that $\rho_0$ is a decreasing function of the weight $\sigma$ of the random component of the perturbation (i.e. of the 'distance' from the optimal perturbation at time 0).

One could argue that in real epidemics most perturbations of the DFE are likely to appear in the form of localized imports of either infected humans or bacteria. Figure 4*b* shows simulations of model (2.1) where the response of the system to different perturbations is measured by means of the output amplification $\alpha_y(t)$. Interestingly, the optimal perturbation at time 0 represents an empirical upper bound for the amplification of point-source perturbations of a stable (yet g-reactive) DFE—not only in the short term (as theoretically expected), but also over longer time scales. Actually, for the parameter setting considered here (characterized, in particular, by a relatively high value of the exposure rate, $\beta$, and a relatively low value of the contamination rate, $\theta$), randomly localized imports of infected human hosts yield transient responses that wane monotonically over time, whereas randomly localized exogenous pathogen loads can lead to a temporary amplification of the system output. These transient responses are topped by the response associated with the optimal perturbation at time 0

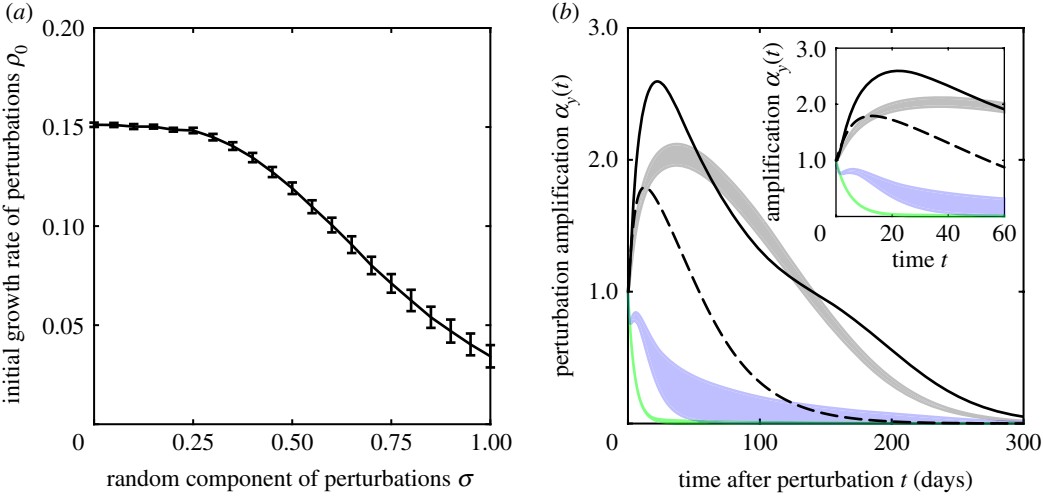

**Figure 4.** Amplification of small perturbations to the DFE over different time scales. (*a*) Initial growth rate $\rho_0$ of spatially distributed random perturbations $\mathbf{w_0}$ to the DFE of model (2.1) evaluated for increasing weights $\sigma$ of their random component (with $\sigma = 0$ corresponding to the optimal perturbation at time 0, and $\sigma = 1$ to a completely random field, see text). Results refer to OCN configurations like those of figure 1*b*. Specifically, shown are the average $\pm 1$ s.d. of the values of $\rho_0$ evaluated over 256 simulations (16 independent perturbations $\times$ 16 OCN replicas) carried out for each value of $\sigma$. (*b*) Long-term amplification $\alpha_y(t) = ||\mathbf{y}(t)||/||\mathbf{y}(0)||$ of the system output for different perturbations to the DFE. The black lines show the effect of the optimal perturbation at time 0 (dashed) and of the asymptotic optimal perturbation (solid). Random point-source perturbations (localized imports of infected humans/pathogens) are shown as green/blue shaded areas (representing envelopes of 100 simulations each), while spatially distributed random perturbations are shown in grey (the shaded area represents, again, an envelope of 100 simulations). The inset provides a closer view of the system response shortly after perturbation. Short- and long-term responses are evaluated via numerical simulations of model (2.1) over the leftmost OCN displayed in figure 1*b*. Parameter values: $\beta = 0.5$, $\theta = 0.01$ (stable, g-reactive DFE). Other parameters as in figure 2*a*. Results obtained for different OCN configurations like those shown in figure 1*a,c* are reported in figure S2, available as electronic supplementary material.

even over relatively long time scales (almost four months in this example). However, model simulations show that the optimal perturbation at time 0 is far from being the most amplified perturbation over longer time scales, actually being taken over by both random, distributed perturbations (e.g. in the form of vector $\mathbf{w_{rand}}$ defined above) and the asymptotic optimal perturbation. The latter, in particular, is indeed found to be among the most amplified perturbations for long time scales, over which linearization could indeed be expected to fail. All these results are robust to changes in the underlying OCN configurations (compare figure 4 with figure S2, in electronic supplementary material).

## 5.4. Analysis of spatial patterns

The spatial signatures of the time-0 optimal perturbation are shown in the left panels of figure 5 (infected components only, but note that bacterial components are qualitatively similar to those shown in the figure) for different parameter combinations, all leading to a stable, g-reactive DFE. In the first two of the four considered parameter sets (panels *a* and *b*; the two parameter combinations are those indicated in figure 2*a*), the backbone of the optimal perturbation at time 0 is localized in a neighbourhood of the network outlet—especially so in the second case, which is characterized by smaller exposure and larger contamination rates. On the contrary, for parameter settings identifying slower hydrologic pathogen transport (*c*) or higher levels of human mobility (*d*), the backbone of the optimal perturbation at time 0 covers larger regions of the landscape, stretching relatively far off upstream the network outlet (which still remains an important focal point in the geography of this critical perturbation), and away from the main course of the stream network. Qualitatively consistent results are found with different OCN configurations (electronic supplementary material, figures S3 and S4).

Interestingly, the spatial structure of the asymptotic optimal perturbation (middle panels of figure 5) seems to be quite robust to changes in the parametrization of hydrologic transport and human mobility, and suggests that the contamination of headwaters or, more generally, of reaches that lie far away from the network outlet may yield relatively large and long-lasting (albeit transient) epidemic waves. The

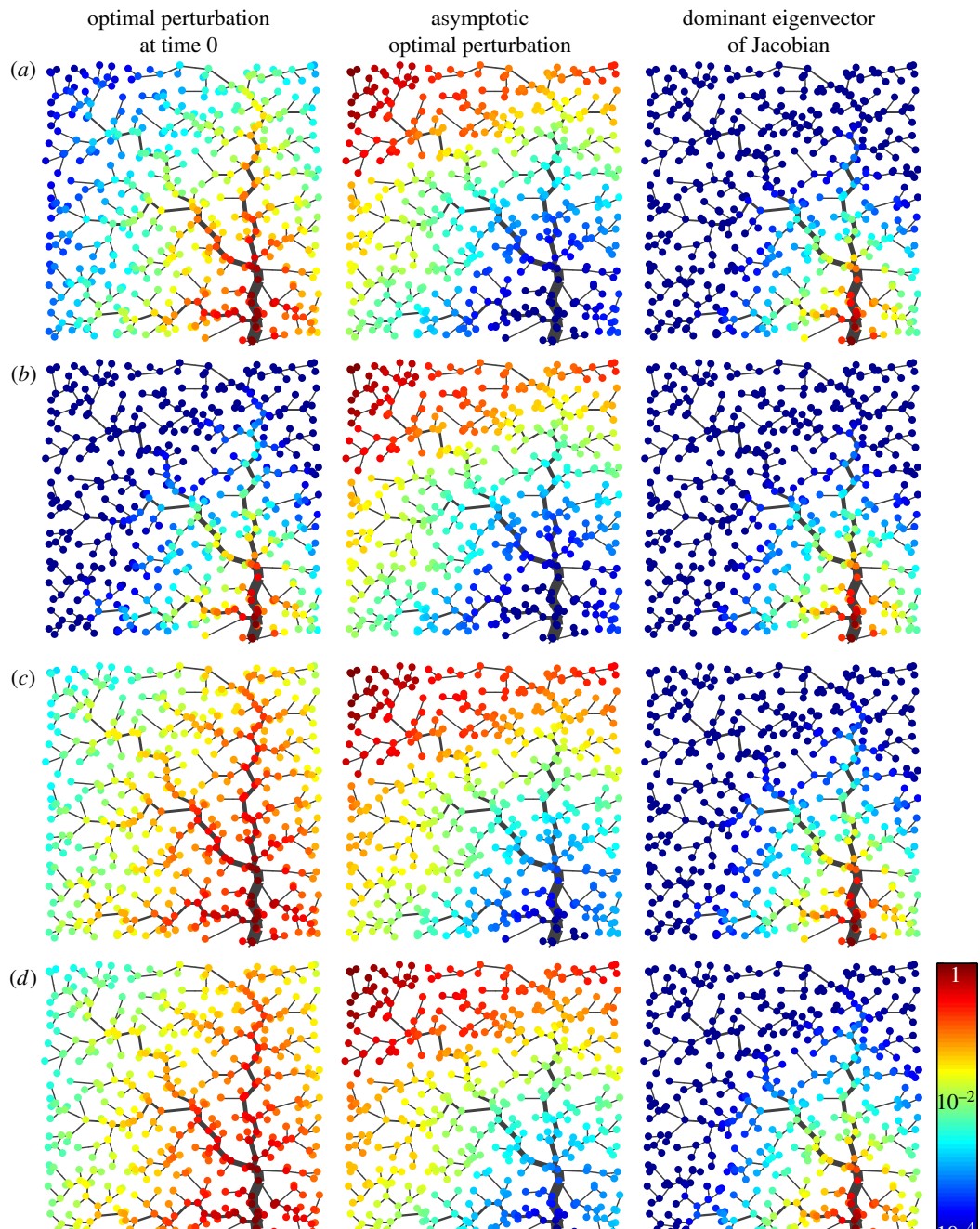

| optimal perturbation at time 0 | asymptotic optimal perturbation | dominant eigenvector of Jacobian |

**Figure 5.** The geography of critical perturbations and of disease spread. Panels show the infected components of the optimal perturbation at time 0 (left), of the asymptotic optimal perturbation (middle) and of the dominant eigenvector of the Jacobian matrix of system (2.1) (right) for different parameter settings. The quantity displayed in each panel is rescaled so that its maximum is equal to 1. All results refer to the leftmost OCN configuration shown in figure 1b. (a) $\beta = 0.5$, $\theta = 0.01$, $l = 1/3$, $m^S = 0.2$, $m^I = 0.05$. (b) As in a, with $\beta = 0.01$ and $\theta = 0.5$. (c) As in a, with $l = 1/6$. (d) As in a, with $m^S = 0.4$ and $m^I = 0.1$. Other parameters as in figure 2a. Results obtained for different OCN configurations like those shown in figure 1a,c are reported in electronic supplementary material, figures S3 and S4.

rationale behind this result is inherently spatial in nature, and can be summarized as follows. In a homogeneous spatial setting, large outbreaks are expected for perturbations that develop into transitory epidemics covering large areal extents in their spatio-temporal evolution; in the absence of sustained transmission and in the presence of directional pathogen movement, this happens to be true for initial conditions that are concentrated farthest from the outlet, as they can indeed generate transient epidemics in which the peak values of local infected prevalence and pathogen concentration

travel downstream as hydrologic transport move pathogens towards the absorbing outlet, thus progressively spanning the whole network.

Finally, the spatial structure of the dominant eigenvector of matrix $\mathbf{J}_0'$ (right panels of figure 5), which describes the geography of the epidemic outbreak as it fades out after the transient phenomena associated with the initial perturbation to the DFE have vanished (appendix A; see also [23,24]), turns out to be pretty robust to changes in model parameters as well, and indicates again the network outlet as a focal point of the spatial epidemic pattern. Interestingly, the perturbation that optimally excites the dominant eigenvector of the Jacobian (the one that dominates in the long run, shown in the middle panels) may bear little resemblance to the eigenvector itself. In fact, as noted by Farrell & Ioannau [38], in generic, non-normal systems (i.e. systems in which the eigenvectors of the associated Jacobian matrix do not form an orthonormal basis), an eigenvector and its biorthogonal may differ greatly.

# 6. Discussion and conclusion

In this work, we have analysed short- and long-term invasion dynamics for waterborne pathogens in realistic riverine landscapes. Specifically, using a well-established, spatially explicit transmission model, we have provided conditions for the DFE of the system (the state in which neither infected human hosts nor free-living pathogens are present in any of the communities included in the region under study) to be g-reactive [29]. Perturbations to a stable, g-reactive DFE may grow in the system output, defined as a linear transformation of the system state that can be chosen to be epidemiologically relevant [34], before eventually fading out. From an epidemiological perspective, a g-reactive DFE may be associated with transient epidemic waves, while an unstable DFE will lead to sustained epidemic dynamics followed by the establishment of endemic transmission.

Understanding the conditions under which short- and/or long-term pathogen transmission is possible in an assigned spatial setting bears important implications for predicting—and possibly controlling—disease spread. Our spatially explicit methods could in fact be used to determine conditions for the occurrence of real-world outbreaks, as well as to assess which communities will be hit with more strength during a waterborne disease epidemic [23]. Results of this kind could thus assist in the planning of interventions aimed to minimize outbreak risk by reducing human exposure and/or contamination, as well as in the management of emergencies in the aftermath of an epidemic outbreak. In this respect, g-reactivity analysis could serve as an ideal companion to asymptotic stability analysis in the definition of a multicriterial decision support framework (e.g. [49]) in which possible trade-offs between short- and long-term objectives can be fully disclosed and clearly examined.

The methods described in this paper can also help evaluate the spatial structure of the perturbations to the DFE that are expected to grow the most in the short run (and, under suitable conditions, in the mid-to-long run as well). Therefore, results from g-reactivity analysis of epidemiological models applied to real case studies could serve as heuristics to define upper bounds of the growth of small perturbations to the DFE, thus providing informed estimates about the maximum size of an outbreak before it goes off. Suggestions from g-reactivity analysis could also be used to optimize the structure of disease surveillance networks (e.g. [50]), specifically to make sure that critical perturbations can be readily identified—and effective interventions set up in a timely manner. For instance, surveillance proved fundamental in the containment of the cholera outbreak that struck Haiti in October 2010 and that has rapidly become one of the largest waterborne disease epidemics of the recent past [51,52].

It has to be remarked, however, that predictions drawn from g-reactivity analysis may prove relevant just shortly after the start of an epidemic in the presence of an unstable DFE. In this case, in fact, linearization is indeed expected to fail rapidly. Needless to say, this scenario is critical from an epidemiological standpoint, because virtually any small perturbation to an unstable DFE can lead to a large outbreak. For waterborne infections, factors contributing to the instability of the DFE are high exposure and contamination risk (say, because of inadequate water provisioning and sanitation infrastructures), the availability of suitable water reservoirs for the thriving of local pathogen populations, and human mobility. History retrospectively shows that all these conditions were unfortunately met in post-earthquake Haiti, where a single contamination event (involving a tributary of the Artibonite River, the largest Haitian fluvial system) from an external source was in fact sufficient to cause a devastating epidemic (e.g. [53–56]). G-reactivity and stability analyses can help understand the role played by different factors in the definition of transmission dynamics over a

range of time scales also in the presence of an unstable DFE, and can be used to increase awareness about the risk of potentially catastrophic outbreaks—a necessary condition to ensure that preventable disasters like the Haiti cholera outbreak do not happen again in the future.

As far as surveillance is concerned, our analysis highlights the neighbourhood of the river network outlet as a critical region for disease spread: on the one hand, in fact, it may constitute the spatial backbone of the optimal perturbation at time 0; on the other, it represents a hotbed for epidemic spread also over longer time scales. These results, together with historical data showing that most outbreaks of cholera (a prototypical example among microparasitic waterborne diseases) originated in coastal regions ([57] and references therein), indicate that the river network outlet should represent a focal point for a surveillance network for waterborne infections and, possibly, for the deployment of healthcare resources. On the other hand, epidemic surveillance and control should not be geographically limited to the outlet region, as the upstream sections of the river network seem to also play an important role, e.g. in the definition of the asymptotic optimal perturbation. Epidemic surveillance and forecasting, design of *ex-ante* and *ex-post* interventions, and resource allocation for transmission control are all crucial objectives of public health policies. Such goals could thus greatly benefit from spatially explicit mathematical modelling.

An important aspect of pathogen transmission that has been neglected in this work is the seasonal variability of model parameters, such as the human exposure and contamination rates, the average pathogen lifespan, or the water reservoir volume [58]. Such temporal variability can result in marked fluctuations of the force of infection, with remarkable consequences for long-term pathogen invasion, as already shown by previous work on the stability properties of the DFE of waterborne disease models in spatially explicit and time-periodic settings [26]. On the other hand, reactivity analysis has never been performed in spatially explicit, time-varying environments. Indeed, the study of the interplay between seasonality and reactivity has started only recently, specifically in spatially implicit ecological systems (e.g. [59]). Because of the local properties of g-reactivity, though, it is safe to predict that seasonal fluctuations of the transmission parameters must play an important—yet possibly nontrivial to unravel—role in the short-term response of the DFE of an epidemiological model to small, time-confined perturbations. A future extension of this work may thus usefully concern the g-reactivity analysis of a seasonal model for waterborne disease dynamics in the framework of Floquet theory (see again [26]).

Although tailored here for waterborne microparasites, spatially explicit tools for g-reactivity and asymptotic stability can be applied to a suite of diverse epidemiological problems. In this respect, the closest application would be the study of g-reactivity for waterborne macroparasitic infections (such as schistosomiasis), which may possibly share similar hydroclimatological and socioeconomic drivers in spite of the different underlying transmission mechanisms [60], and for which spatially explicit modelling approaches are becoming increasingly available (e.g. [25,61–64]). Moreover, future studies may also focus on spatially explicit models for diseases with different infection pathways, including human-to-human, airborne, faecal-oral, sexual and vector-borne. Properly informed with the spatial mechanisms relevant to each transmission route, g-reactivity and stability analyses could help ecologists and epidemiologists better understand how pathogen transport and human mobility interact with local transmission mechanisms in shaping short- and long-term invasion dynamics, with important consequences for the effectiveness of infectious disease control.

Data accessibility. This work makes no use of original data. Sections 3 and 4 in the manuscript provide detailed instructions to reproduce the results presented in §5, where the model described in §2 is applied to theoretical riverine landscapes produced following procedures that are fully described elsewhere (references in §5). All analyses have been performed using numerical algorithms that are routinely implemented in standard scientific computing software. Specifically, MATLAB$^{TM}$ version R2017a has been used for this work. A sample MATLAB$^{TM}$ implementation of the instructions needed to evaluate the stability and g-reactivity properties of model (2.1) with the output transformation defined in (3.2) is provided in appendix C (electronic supplementary material).

Authors' contributions. L.M. conceived and coordinated the study, performed the numerical analyses and drafted the first version of the manuscript. R.C. contributed to drafting the manuscript. E.B. participated in numerical analyses. A.R. and M.G. contributed to conceiving and coordinating the study. All authors participated in the design of the study and the revision of the manuscript, and gave final approval for publication.

Competing interests. The authors have no competing interests.

Funding. L.M., R.C. and M.G. acknowledge support from Politecnico di Milano. L.M. and R.C. were also supported by Politecnico di Milano through the Polisocial Award programme, project MASTR-SLS. A.R. acknowledges funding from the ERC Advanced Grant RINEC 22761 'River networks as ecological corridors for species, populations, and

pathogens of waterborne disease' and from the Swiss National Science Foundation project 'Optimal control of intervention strategies for waterborne disease epidemics' (SNSF grant no. 200021_172578).

Acknowledgements. The authors wish to thank three anonymous reviewers for their useful comments on the original version of the manuscript.

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
