## [Reviewer comments · Royal Society Open Science]

Review History

RSOS-181517.R0 (Original submission)

Review form: Reviewer 1

Is the manuscript scientifically sound in its present form?

Yes

Are the interpretations and conclusions justified by the results?

Yes

Is the language acceptable?

Yes

Is it clear how to access all supporting data?

No

Do you have any ethical concerns with this paper?

No

Have you any concerns about statistical analyses in this paper?

No

Recommendation?

Accept with minor revision (please list in comments)

Comments to the Author(s)

This work deals with the mathematical analysis of stability conditions of the disease-free equilibrium in an eco-epidemiological model that describes the spread of a waterborne disease through a networked system of populations.

In particular, the authors analyse the conditions under which a small perturbation of the system (for instance the introduction of few infected individuals) can lead to transient epidemics or long term endemic conditions.

The analysis is based on the concept of g -reactivity which has been introduced by the authors in previous works.

Overall, the paper presents novel results that are sound and important for the scientific community of mathematical ecologists and epidemiologists.

The g -reactivity framework is shown to be a powerful tool to understand the dynamics of spatially explicit epidemic models.

The paper is definitively suitable for publication in the Royal Society Open Science.

I have only minor suggestions for revisions:

There is a distinction between the model described by equations (1) with and without the addition of the OCN and the human mobility network. However, I found difficult to understand where these two additional layers are included and where they are not. Is all the analysis of sections 3-4 derived without considering the spatial networks above? I would suggest clarifying this point with a few lines.

I would suggest adding a table summarising all the parameters of the model and the notation used.

Does the assumption on the mobility model used (in this case the gravity model) affect the results? Could you please add a comment on this?

Line 270, I guess there is a typo: $Q_{ii}=0$.

Code availability. Even if the authors state that sections 3 and 4 provide detailed instructions to reproduce the results of section 5, I am sure there are several technical details in the implementation of the model that are not obvious for a non-expert reader. It would be greatly beneficial for researchers working on closely related problems to share the code used for this work in a public repository. I am sure that sharing the code would also increase significantly the impact of the manuscript.

Review form: Reviewer 2**Is the manuscript scientifically sound in its present form?**

Yes

Are the interpretations and conclusions justified by the results?

Yes

Is the language acceptable?

Yes

Is it clear how to access all supporting data?

No

Do you have any ethical concerns with this paper?

No

Have you any concerns about statistical analyses in this paper?

No

Recommendation?

Accept as is

Comments to the Author(s)

The manuscript extends a previously published analytical approach to include complex spatial models. This approach now allows one to identify conditions under which transient epidemics could occur and to explore the spatial structure of the initial conditions that can generate such epidemics.

The methods for analyzing the model are described well in Sections 3 and 4, the metapopulation model itself is quite complex, with 500 populations linked by a river-like network. It would be difficult for a reader to replicate the model and analysis. The methods here would be much more useful if they were accompanied by code to generate the models and to analyze them, though I am not sure this would be required by the journal. Journal policy is to share data and code supporting the manuscript results (see the "Data sharing" section at <https://royalsocietypublishing.org/rsos/for-reviewers>).

Minor comments:

On first read, I had some trouble understanding the connection between reactivity in the Introduction (page 4) and g -reactivity (page 5). I think defining g -reactivity early on page 5 would help (it is now first defined in Section 3 on page 8).

Line 388 (page 15) describes the long-term persistence of epidemics. Does that mean that a long-term intervention targeting communities near the river outlet such as routine vaccination could be most effective at preventing persistence?

Line 432 (page 19) notes that cholera outbreaks tend to start in coastal regions. It is interesting that the asymptotic optimal perturbation to the model is not near the river outlet but in the headwaters (Section 5.4).

I am having some trouble understanding the paragraph in the Discussion beginning on line 438 (page 20). The paragraph seems to be about the unstable disease-free equilibrium (DFE) and not about the g -reactive stable DFE, so the relationship between Haiti and the analytical framework is not clear.

Review form: Reviewer 3

Is the manuscript scientifically sound in its present form?

Yes

Are the interpretations and conclusions justified by the results?

No

Is the language acceptable?

Yes

Is it clear how to access all supporting data?

No

Do you have any ethical concerns with this paper?

No

Have you any concerns about statistical analyses in this paper?

No

Recommendation?

Accept with minor revision (please list in comments)

Comments to the Author(s)

The manuscript by Mari et al describes a network model that they used to evaluate the roles of human mobility and water-borne transport in propagating and sustaining epidemics of water-borne pathogens. The analysis appears to be technically sound (although I was not able to follow all of the math) and is novel, although the authors overstate the applicability of their findings and make a number of simplifying assumptions that make the work largely theoretical.

My main concern is that the authors repeatedly overstate the “realism” of their model and its applicability to real-world epidemic surveillance and control. The model does not appear to be based on or fitted to data for any particular pathogen or population. Furthermore, the model is based on the classic SIR paradigm of a fully immunizing infection, and yet I do not believe there is any water-borne pathogen that confers life-long immunity to reinfection. While an SIR model structure may be useful for examining the invasion dynamics of a pathogen that confers temporary immunity to reinfection in a fully susceptible population (as is reasonable for modeling the initial introduction of cholera to Haiti, for example), here the authors are interested in exploring the equilibrium dynamics of the system and occurrence of transient epidemics and/or endemicity over longer timescales. As such, they should make explicit the assumption of lifelong immunity and discuss the potential implications and limitations of this assumption. Lastly, the manuscript is written for a highly technical audience, and would not be accessible to an audience that could actually translate the findings into actual practical use, e.g. the planning of interventions or design of surveillance systems, as suggested on p. 19. It may be more useful to discuss how their approach might be incorporated into or used to inform models fitted to real data that could actually be validated in some way.

Minor comments:

p. 3, line 20: Replace “in general” with “generally”.

p. 3, lines 24-30: Rotavirus, which is the leading cause of mortality from diarrhea in children, is generally not considered to be primarily waterborne. It is still a leading cause of diarrhea in

developed countries with improved water and sanitation (prior to vaccine introduction), and epidemics tend to peak during the dry season in most countries throughout the world. Therefore, it should not be mentioned as an example of a waterborne infection, nor should it be implied that most diarrheal disease is “clearly” attributable to unsafe water.

p. 7, lines 126-130: I believe it would be more accurate to refer to the parameter m_i as the fraction of time that mobile individuals living in i spend outside of their own community. Otherwise, it sounds as though this fraction of (discrete) individuals are always elsewhere (in communities other than i), which does not make sense.

p. 9, line 189: Replace “work” with “worth”.

p. 19, line 425: Replace “timely set up” with “set up in a timely manner”.

p. 19, line 426: Replace “stroke” with “struck”.

p. 19, line 436: Replace “forecast” with “forecasting”.

Data accessibility: The code for the model should be shared, either as supplementary material or in a repository such as github.

Decision letter (RSOS-181517.R0)

21-Feb-2019

Dear Dr Mari

On behalf of the Editors, I am pleased to inform you that your Manuscript RSOS-181517 entitled "Conditions for transient epidemics of waterborne disease in spatially explicit systems" has been accepted for publication in Royal Society Open Science subject to minor revision in accordance with the referee suggestions. Please find the referees' comments at the end of this email.

The reviewers and handling editors have recommended publication, but also suggest some minor revisions to your manuscript. Therefore, I invite you to respond to the comments and revise your manuscript.

- Ethics statement

- Data accessibility

It is a condition of publication that all supporting data are made available either as supplementary information or preferably in a suitable permanent repository. The data accessibility section should state where the article's supporting data can be accessed. This section should also include details, where possible of where to access other relevant research materials such as statistical tools, protocols, software etc can be accessed. If the data has been deposited in an external repository this section should list the database, accession number and link to the DOI for all data from the article that has been made publicly available. Data sets that have been

deposited in an external repository and have a DOI should also be appropriately cited in the manuscript and included in the reference list.

If you wish to submit your supporting data or code to Dryad (<http://datadryad.org/>), or modify your current submission to dryad, please use the following link:
<http://datadryad.org/submit?journalID=RSOS&manu=RSOS-181517>

- **Competing interests**

- **Authors' contributions**

- **Acknowledgements**

- **Funding statement**

Because the schedule for publication is very tight, it is a condition of publication that you submit the revised version of your manuscript before 02-Mar-2019. Please note that the revision deadline will expire at 00.00am on this date. If you do not think you will be able to meet this date please let me know immediately.

When submitting your revised manuscript, you will be able to respond to the comments made by

the referees and upload a file "Response to Referees" in "Section 6 - File Upload". You can use this to document any changes you make to the original manuscript. In order to expedite the processing of the revised manuscript, please be as specific as possible in your response to the referees. We strongly recommend uploading two versions of your revised manuscript:

Kind regards,
Andrew Dunn

Royal Society Open Science Editorial Office
 Royal Society Open Science
 openscience@royalsociety.org

on behalf of Professor Anotida Madzvamuse (Associate Editor) and Jon Blundy (Subject Editor)
 openscience@royalsociety.org

Associate Editor Comments to Author (Professor Anotida Madzvamuse):

Associate Editor: 1

Comments to the Author:

My recommendation is to accept the manuscript subject to authors addressing the minor comments raised.

Associate Editor: 2

Comments to the Author:

I recommend peer review of the manuscript

Reviewer comments to Author:

Reviewer: 1

Comments to the Author(s)

This work deals with the mathematical analysis of stability conditions of the disease-free equilibrium in an eco-epidemiological model that describes the spread of a waterborne disease through a networked system of populations.

In particular, the authors analyse the conditions under which a small perturbation of the system (for instance the introduction of few infected individuals) can lead to transient epidemics or long term endemic conditions.

The analysis is based on the concept of g -reactivity which has been introduced by the authors in previous works.

Overall, the paper presents novel results that are sound and important for the scientific community of mathematical ecologists and epidemiologists.

The g -reactivity framework is shown to be a powerful tool to understand the dynamics of spatially explicit epidemic models.

The paper is definitively suitable for publication in the Royal Society Open Science.

I have only minor suggestions for revisions:

There is a distinction between the model described by equations (1) with and without the addition of the OCN and the human mobility network. However, I found difficult to understand where these two additional layers are included and where they are not. Is all the analysis of sections 3-4 derived without considering the spatial networks above? I would suggest clarifying this point with a few lines.

I would suggest adding a table summarising all the parameters of the model and the notation used.

Does the assumption on the mobility model used (in this case the gravity model) affect the results? Could you please add a comment on this?

Line 270, I guess there is a typo: $Q_{ii}=0$.

Code availability. Even if the authors state that sections 3 and 4 provide detailed instructions to reproduce the results of section 5, I am sure there are several technical details in the implementation of the model that are not obvious for a non-expert reader. It would be greatly beneficial for researchers working on closely related problems to share the code used for this work in a public repository. I am sure that sharing the code would also increase significantly the impact of the manuscript.

Reviewer: 2

Comments to the Author(s)

The manuscript extends a previously published analytical approach to include complex spatial models. This approach now allows one to identify conditions under which transient epidemics could occur and to explore the spatial structure of the initial conditions that can generate such epidemics.

The methods for analyzing the model are described well in Sections 3 and 4, the metapopulation model itself is quite complex, with 500 populations linked by a river-like network. It would be difficult for a reader to replicate the model and analysis. The methods here would be much more useful if they were accompanied by code to generate the models and to analyze them, though I am not sure this would be required by the journal. Journal policy is to share data and code supporting the manuscript results (see the "Data sharing" section at <https://royalsocietypublishing.org/rsos/for-reviewers>).

Minor comments:

On first read, I had some trouble understanding the connection between reactivity in the Introduction (page 4) and g -reactivity (page 5). I think defining g -reactivity early on page 5 would help (it is now first defined in Section 3 on page 8).

Line 388 (page 15) describes the long-term persistence of epidemics. Does that mean that a long-term intervention targeting communities near the river outlet such as routine vaccination could be most effective at preventing persistence?

Line 432 (page 19) notes that cholera outbreaks tend to start in coastal regions. It is interesting that the asymptotic optimal perturbation to the model is not near the river outlet but in the headwaters (Section 5.4).

I am having some trouble understanding the paragraph in the Discussion beginning on line 438 (page 20). The paragraph seems to be about the unstable disease-free equilibrium (DFE) and not about the g -reactive stable DFE, so the relationship between Haiti and the analytical framework is not clear.

Reviewer: 3

Comments to the Author(s)

The manuscript by Mari et al describes a network model that they used to evaluate the roles of human mobility and water-borne transport in propagating and sustaining epidemics of water-borne pathogens. The analysis appears to be technically sound (although I was not able to follow all of the math) and is novel, although the authors overstate the applicability of their findings and make a number of simplifying assumptions that make the work largely theoretical.

My main concern is that the authors repeatedly overstate the "realism" of their model and its

applicability to real-world epidemic surveillance and control. The model does not appear to be based on or fitted to data for any particular pathogen or population. Furthermore, the model is based on the classic SIR paradigm of a fully immunizing infection, and yet I do not believe there is any water-borne pathogen that confers life-long immunity to reinfection. While an SIR model structure may be useful for examining the invasion dynamics of a pathogen that confers temporary immunity to reinfection in a fully susceptible population (as is reasonable for modeling the initial introduction of cholera to Haiti, for example), here the authors are interested in exploring the equilibrium dynamics of the system and occurrence of transient epidemics and/or endemicity over longer timescales. As such, they should make explicit the assumption of lifelong immunity and discuss the potential implications and limitations of this assumption. Lastly, the manuscript is written for a highly technical audience, and would not be accessible to an audience that could actually translate the findings into actual practical use, e.g. the planning of interventions or design of surveillance systems, as suggested on p. 19. It may be more useful to discuss how their approach might be incorporated into or used to inform models fitted to real data that could actually be validated in some way.

Minor comments:

p. 3, line 20: Replace “in general” with “generally”.

p. 3, lines 24-30: Rotavirus, which is the leading cause of mortality from diarrhea in children, is generally not considered to be primarily waterborne. It is still a leading cause of diarrhea in developed countries with improved water and sanitation (prior to vaccine introduction), and epidemics tend to peak during the dry season in most countries throughout the world. Therefore, it should not be mentioned as an example of a waterborne infection, nor should it be implied that most diarrheal disease is “clearly” attributable to unsafe water.

p. 7, lines 126-130: I believe it would be more accurate to refer to the parameter m_i as the fraction of time that mobile individuals living in i spend outside of their own community. Otherwise, it sounds as though this fraction of (discrete) individuals are always elsewhere (in communities other than i), which does not make sense.

p. 9, line 189: Replace “work” with “worth”.

p. 19, line 425: Replace “timely set up” with “set up in a timely manner”.

p. 19, line 426: Replace “stroke” with “struck”.

p. 19, line 436: Replace “forecast” with “forecasting”.

Data accessibility: The code for the model should be shared, either as supplementary material or in a repository such as github.

Author's Response to Decision Letter for (RSOS-181517.R0)

See Appendix A.

Decision letter (RSOS-181517.R1)

12-Mar-2019

Dear Dr Mari,

I am pleased to inform you that your manuscript entitled "Conditions for transient epidemics of waterborne disease in spatially explicit systems" is now accepted for publication in Royal Society Open Science.

on behalf of Professor Anotida Madzvamuse (Associate Editor) and Professor Jon Blundy (Subject Editor)
openscience@royalsociety.org

Appendix A

Conditions for transient epidemics of waterborne disease in spatially explicit systems

— Manuscript ID RSOS-181517 —

Point-by-point **Authors' answers** to Referees' comments

Lorenzo Mari, Renato Casagrandi, Enrico Bertuzzo, Andrea Rinaldo, Marino Gatto

March 7, 2019

1 Referee #1

[R1.1] This work deals with the mathematical analysis of stability conditions of the disease-free equilibrium in an eco-epidemiological model that describes the spread of a waterborne disease through a networked system of populations. In particular, the authors analyse the conditions under which a small perturbation of the system (for instance the introduction of few infected individuals) can lead to transient epidemics or long term endemic conditions. The analysis is based on the concept of g-reactivity which has been introduced by the authors in previous works. Overall, the paper presents novel results that are sound and important for the scientific community of mathematical ecologists and epidemiologists. The g-reactivity framework is shown to be a powerful tool to understand the dynamics of spatially explicit epidemic models. The paper is definitively suitable for publication in the Royal Society Open Science. I have only minor suggestions for revisions.

We thank this referee for her/his excellent summary and positive assessment of our work.

[R1.2] There is a distinction between the model described by equations (1) with and without the addition of the OCN and the human mobility network. However, I found difficult to understand where these two additional layers are included and where they are not. Is all the analysis of sections 3-4 derived without considering the spatial networks above? I would suggest clarifying this point with a few lines.

The disease transmission model is presented in equations (1) in its most general formulation, i.e. accounting for both local and spatial processes. All of the analyses described in sections 3 and 4, as well as the numerical results presented in section 5, refer to this formulation. Human mobility and hydrological transport of pathogens can be 'shut down' by setting $m_i^S = m_i^I = 0$ and $l_i = 0$ for all i 's, respectively. This actually produces a set of disconnected local models, which is the case shown as thin gray lines in Figure 2. We

acknowledge that the relationship between the full spatial model and its spatially-implicit counterpart was mentioned only in the caption of Figure 2. We have therefore added a few lines of explanation in the main text about this important point (p. 8, ll. 159–163 in the revised manuscript). Thanks for pointing out this potential source of misunderstandings.

[R1.3] I would suggest adding a table summarising all the parameters of the model and the notation used.

A table summarizing all the variables and parameters of the model has been added (Table 1, p. 32). Thanks for the suggestion.

[R1.4] Does the assumption on the mobility model used (in this case the gravity model) affect the results? Could you please add a comment on this?

A different mobility model could be easily accommodated within the formalism of model (1), provided that human mobility may be expressed in terms of movement probability (as quantified by the parameters m_i^S and m_i^I) and trip distribution (e.g. in the form of an origin-destination matrix, as quantified by matrix \mathbf{Q}). Nothing would change in this case from the perspective of the analytical derivation of the conditions for pathogen endemicity or short-term outbreaks. Clearly, however, numerical results might differ considerably, as also demonstrated by the numerical experiment shown in Figure 3b, where wide ranges of variation for the parameters of the gravity model (corresponding to different human mobility patterns) have been considered. We have revised our manuscript to include these comment (p. 14, ll. 292–294).

[R1.5] Line 270, I guess there is a typo: $Q_{ii} = 0$.

Corrected, thanks (p. 14 l. 289).

[R1.6] Code availability. Even if the authors state that sections 3 and 4 provide detailed instructions to reproduce the results of section 5, I am sure there are several technical details in the implementation of the model that are not obvious for a non-expert reader. It would be greatly beneficial for researchers working on closely related problems to share the code used for this work in a public repository. I am sure that sharing the code would also increase significantly the impact of the manuscript.

Thanks for raising this point. Following this Reviewer's suggestion (and similar ones by the other two Referees as well, see below) we have included a MATLAB implementation of the instructions required to evaluate epidemicity and endemicity conditions, as well as to perform numerical simulations of our model, in a new section of the Electronic Supplementary Material (ESM; see Appendix C, pp. 7–11).

2 Referee #2

[R2.1] The manuscript extends a previously published analytical approach to include complex spatial models. This approach now allows one to identify conditions under which transient epidemics could occur and to explore the spatial structure of the initial conditions that can generate such epidemics. The methods for analyzing the model are described well in Sections 3 and 4, the metapopulation model itself is quite complex, with 500 populations linked by a river-like network. It would be difficult for a reader to replicate the model and analysis. The methods here would be much more useful if they were accompanied by code to generate the models and to analyze them, though I am not sure this would be required by the journal. Journal policy is to share data and code supporting the manuscript results (see the “Data sharing” section at <https://royalsocietypublishing.org/rsos/for-reviewers>).

We understand the concern raised by this Reviewer, and we agree that implementing from scratch the methodology described in this work might be nontrivial for some readers. Following also similar suggestions from the other Reviewers, a MATLAB implementation of stability/g-reactivity analysis of the DFE of model (1) is provided in a new dedicated section of the ESM (Appendix C, pp. 7–11). Thanks for raising this important point.

[R2.2] Minor comments: On first read, I had some trouble understanding the connection between reactivity in the Introduction (page 4) and g-reactivity (page 5). I think defining g-reactivity early on page 5 would help (it is now first defined in Section 3 on page 8).

Thanks for the remark. A short definition of g-reactivity is now provided when the concept is first introduced (p. 5, ll. 78–80).

[R2.3] Line 388 (page 15) describes the long-term persistence of epidemics. Does that mean that a long-term intervention targeting communities near the river outlet such as routine vaccination could be most effective at preventing persistence?

This is an interesting observation. For sure, the fact that the outlet and the river stretches closest to it are where most of the cases are expected over relatively long time scales suggests that this region requires intensified surveillance (as already noted in the original manuscript; see p. 19, ll. 429–435) and likely qualifies as an intervention hotspot (as now highlighted in the revised manuscript; see p. 21, ll. 472–473). However, we refrain from stating that interventions geographically targeted to the outlet region would be optimal for the prevention of sustained disease transmission, because drawing such a conclusion would require an in-depth exploration of different spatial prioritization options for the deployment of health-care resources – something, we feel, that goes beyond the scope of this work.

[R2.4] Line 432 (page 19) notes that cholera outbreaks tend to start in coastal regions. It is interesting that the asymptotic optimal perturbation to the model is not near the river outlet but in the headwaters (Section 5.4).

Thanks for the annotation. We have added a comment to emphasize the role possibly played by the upstream sections of the river network in the development of transient outbreaks (p. 21, ll. 473–475).

[R2.5] I am having some trouble understanding the paragraph in the Discussion beginning on line 438 (page 20). The paragraph seems to be about the unstable disease-free equilibrium (DFE) and not about the g -reactive stable DFE, so the relationship between Haiti and the analytical framework is not clear.

Indeed, that paragraph is mostly concerned with an intrinsic limitation of the g -reactivity framework in the presence of an asymptotically unstable DFE (or any unstable equilibrium point of a generic system of ordinary differential equations for that matter). In that case, in fact, the exponential divergence of the system trajectories from the unstable equilibrium would cause linearization (a staple of both linear stability and g -reactivity analyses) to fail rapidly, thus making any predictions drawn from g -reactivity less relevant for the problem under study. However, we maintain that performing g -reactivity and stability analyses can be useful to better understand the role played by different risk factors in the definition of transmission dynamics over a range of time scales. We have slightly rephrased the paragraph (p. 21, ll. 461–465) to improve its readability.

3 Referee #3

[R3.1] The manuscript by Mari et al describes a network model that they used to evaluate the roles of human mobility and water-borne transport in propagating and sustaining epidemics of water-borne pathogens. The analysis appears to be technically sound (although I was not able to follow all of the math) and is novel, although the authors overstate the applicability of their findings and make a number of simplifying assumptions that make the work largely theoretical.

We agree with this Reviewer that the results presented in this work are mainly theoretical in their spirit. Let us just note, however, that in spite of all its limitations, model (1) has been quite successfully applied to actual epidemic outbreaks, as also pointed out in the original version of the manuscript (p. 6, ll. 107–111). We are thus confident that at least some of the results presented in this work may apply as well to real-world case studies.

[R3.2] My main concern is that the authors repeatedly overstate the “realism” of their model and its applicability to real-world epidemic surveillance and control. The model does not appear to be based

on or fitted to data for any particular pathogen or population.

We respectfully disagree on this point. The biological parameters of model (1) are in fact taken from a study (Bertuzzo et al., 2016) where the model was calibrated against actual case reports from the cholera epidemic that struck Haiti at the end of 2010, as explained in the caption of Figure 2 (p. 30, l. 690 of the original manuscript). We have revised the manuscript to make clear that, despite its apparent generality, model (1) is actually best suited to describe cholera transmission dynamics (p. 5, ll. 97–98; p. 6, ll. 111–113). Also, the landscape structure used for the numerical examples described in this work has been quite often analyzed in other epidemiological applications for its resemblance to actual fluvial systems. A comment on this point has been added to the revised version of the manuscript (p. 13, ll. 264–266).

[R3.3] Furthermore, the model is based on the classic SIR paradigm of a fully immunizing infection, and yet I do not believe there is any water-borne pathogen that confers life-long immunity to reinfection. While an SIR model structure may be useful for examining the invasion dynamics of a pathogen that confers temporary immunity to reinfection in a fully susceptible population (as is reasonable for modeling the initial introduction of cholera to Haiti, for example), here the authors are interested in exploring the equilibrium dynamics of the system and occurrence of transient epidemics and/or endemicity over longer timescales. As such, they should make explicit the assumption of lifelong immunity and discuss the potential implications and limitations of this assumption.

The g-reactivity and stability thresholds analyzed in this work refer to the DFE of model (1). In a disease-free condition, by definition, the whole host population is assumed to be in a susceptible state. For this reason, deciding whether loss of immunity from recovered individual should be included in the model is immaterial. This was briefly mentioned in the original version of the manuscript at p. 7, ll. 135–138. That comment has now been emphasized as suggested by this Referee (p. 7, ll. 139–142). We also note that recent estimates from the Haiti cholera epidemic place the duration of acquired immunity in the range 2.3–3.0 years (Pasetto et al., 2018). As such, immunity loss is unlikely to influence the temporal dynamics of disease transmission over relatively short time scales, e.g. like those used for the simulations of Figure 4. A comment on this point has been added to the revised version of the manuscript (pp. 7–8, ll. 142–144).

[R3.4] Lastly, the manuscript is written for a highly technical audience, and would not be accessible to an audience that could actually translate the findings into actual practical use, e.g. the planning of interventions or design of surveillance systems, as suggested on p. 19. It may be more useful to discuss how their approach might be incorporated into or used to inform models fitted to real data that could actually be validated in some way.

We agree that the manuscript is pretty technical, especially in sections 3 and 4. We believe, however, that the numerical examples shown in section 5 may appeal a wider audience, and special care has been

devoted to make sure that their description (along with the accompanying Discussion section) is accessible to readers with a less quantitative background. We anticipate that, in the same spirit, we have also included in the revision a simple MATLAB script that shows how to perform epidemicity and endemicity analysis with our model in a step-by-step fashion (see comment R3.12 below and similar remarks made by the other Reviewers). For the comment on the applicability of our framework to a model that can be fitted to real data, please refer to the answer to point R3.2 above.

[R3.5] Minor comments: p. 3, line 20: Replace “in general” with “generally”.

Done, thanks (p. 3, l. 20).

[R3.6] p. 3, lines 24-30: Rotavirus, which is the leading cause of mortality from diarrhea in children, is generally not considered to be primarily waterborne. It is still a leading cause of diarrhea in developed countries with improved water and sanitation (prior to vaccine introduction), and epidemics tend to peak during the dry season in most countries throughout the world. Therefore, it should not be mentioned as an example of a waterborne infection, nor should it be implied that most diarrheal disease is “clearly” attributable to unsafe water.

Thanks for the comment. We have revised the paragraph at p. 3, ll. 23–29.

[R3.7] p. 7, lines 126-130: I believe it would be more accurate to refer to the parameter m_i as the fraction of time that mobile individuals living in i spend outside of their own community. Otherwise, it sounds as though this fraction of (discrete) individuals are always elsewhere (in communities other than i), which does not make sense.

Agreed. We have modified the description of the parameter accordingly (p. 7, ll. 130–132; see also p. 8, ll. 150–152, and p. 11, ll. 210–212). Thanks for raising this point.

[R3.8] p. 9, line 189: Replace “work” with “worth”.

Done, thanks (p. 10, l. 207).

[R3.9] p. 19, line 425: Replace “timely set up” with “set up in a timely manner”.

Done, thanks (p. 20, l. 446–447).

[R3.10] p. 19, line 426: Replace “stroke” with “struck”.

Done, thanks (p. 20, l. 448).

[R3.11] p. 19, line 436: Replace “forecast” with “forecasting”.

Done, thanks (p. 21, l. 476).

[R3.12] Data accessibility: The code for the model should be shared, either as supplementary material or in a repository such as github.

Following this comments and similar ones made by the other two Reviewers, we have included a MATLAB implementation of the instructions required to evaluate the stability and g-reactivity properties of the model described in this work in a new section of the ESM (Appendix C, pp. 7–11). Thanks for this sensible suggestion.

In conclusion, we wish to thank the editors for giving us the opportunity to revise our manuscript, and the three reviewers for their useful comments that indeed helped us improve our work.

References

- Bertuzzo, E., Finger, F., Mari, L., Gatto, M., and Rinaldo, A. (2016). On the probability of extinction of the Haiti cholera epidemic. *Stochastic Environmental Research and Risk Assessment*, 30:2043–2055.
- Pasetto, D., Finger, F., Camacho, A., Grandesso, F., Cohuet, S., Lemaitre, J. C., Azman, A. S., Luquero, F. J., Bertuzzo, E., and Rinaldo, A. (2018). Near real-time forecasting for cholera decision making in Haiti after Hurricane Matthew. *PLoS Computational Biology*, 14:e1006127.